# Predicting Lifestyle from Positive Selection Data and Genome Properties in Oomycetes

**DOI:** 10.3390/pathogens10070807

**Published:** 2021-06-25

**Authors:** Daniel Gómez-Pérez, Eric Kemen

**Affiliations:** Center for Plant Molecular Biology (ZMBP), University of Tübingen, 72074 Tübingen, Germany; daniel.gomez-perez@uni-tuebingen.de

**Keywords:** oomycetes, lifestyle, evolution

## Abstract

As evidenced in parasitism, host and niche shifts are a source of genomic and phenotypic diversification. Exemplary is a reduction in the core metabolism as parasites adapt to a particular host, while the accessory genome often maintains a high degree of diversification. However, selective pressures acting on the genome of organisms that have undergone recent lifestyle or host changes have not been fully investigated. Here, we developed a comparative genomics approach to study underlying adaptive trends in oomycetes, a eukaryotic phylum with a wide and diverse range of economically important plant and animal parasitic lifestyles. Our analysis reveals converging evolution on biological processes for oomycetes that have similar lifestyles. Moreover, we find that certain functions, in particular carbohydrate metabolism, transport, and signaling, are important for host and environmental adaptation in oomycetes. Given the high correlation between lifestyle and genome properties in our oomycete dataset, together with the known convergent evolution of fungal and oomycete genomes, we developed a model that predicts plant pathogenic lifestyles with high accuracy based on functional annotations. These insights into how selective pressures correlate with lifestyle may be crucial to better understand host/lifestyle shifts and their impact on the genome.

## 1. Introduction

The adaptation of organisms as they evolve to occupy different niches or adopt different lifestyles is reflected in their genome. Expansion or contraction of gene families has been cited as a general mechanism for such adaptations [1,2]. Expansions arise mainly from gene duplication and, in some cases, from acquisition via horizontal gene transfer, whereas gene loss can happen by accumulation of loss-of-function mutations through genetic drift [3,4,5]. Fundamentally, both of these processes are driven by adaptive evolution, whereby beneficial mutations are selected for and deleterious ones removed from the gene pool, ultimately leading to phenotypic diversification [6]. More concretely, trends in the evolution of coding genes can be studied by measuring the ratio of non-synonymous (dN) to synonymous (dS) amino acid rates in the comparison to closely related sequences, usually represented as ω [7]. A ratio higher than one (dN/dS=ω>1) implies positive selection and thus functional diversification, while a ratio lower than one (dN/dS=ω<1) indicates the presence of purifying selection and thus a tighter constraint for the diversification of the gene sequence. Most genes in an organism are under strong purifying selection, as a change in a key amino acid of a protein would have a detrimental effect [8]. However, a small portion of them, those that have been subject to recent diversification, show signs of an increased nonsynonymous mutation rate. Codon models that take into account statistical rate variations are commonly used in comparative genomic studies [9]. When performed on related organisms that have different lifestyles and hosts, the study of positively selected genes together with their functional annotation illustrates which gene functions played important roles in the adaptation process.

Oomycetes are eukaryotic organisms belonging, together with diatoms and brown algae, to the group of stramenopiles [10,11]. Since their origin from a marine autotrophic common ancestor around 400 million years ago, oomycetes have adapted to multiple environments and lifestyles, and many of them are economically impactful plant and animal parasites [12,13,14]. Therefore, they represent a relevant and appropriate system to study the genetic impact of lifestyle and host adaptation on genetically close genomes. Four phylogenetic families, representative of oomycete’s large diversity, are the target of most current research efforts: Albuginaceae, Peronosporaceae, Saprolegniaceae, and Pythiaceae. The Albuginaceae and most Peronosporaceae independently evolved the ability to survive exclusively on living host material, also known as obligate biotrophy [15]. However, some Peronosporaceae, particularly in the *Phytophthora* genus, are hemibiotrophs, i.e., they display an initial biotrophic phase followed by a necrotrophic one, during which they feed on the decaying living matter of their host [16]. Additionally in the Peronosporaceae, the early divergent clade of *Globisporangium* consists of plant necrotrophs previously classified as *Pythiaceae*. All Albuginaceae, Peronosporaceae, and most Pythiaceae are plant parasitic organisms [17]. On the contrary, most Saprolegniaceae are capable of infecting animals, with few exceptions including plant-causing root rot *Aphanomyces* and the free-living saprophyte *Thraustotheca clavata*, which does not need a host at any point in its life cycle [18,19,20].

Obligate biotrophs have a considerably reduced primary metabolism. Comparative genome studies have reported a significant and convergent loss of the enzymatic arsenal in independent lineages of the oomycetes following this lifestyle [21]. The picture is not so clear for the heterotrophs and their adaptation to different hosts. *Pythium insidiosum*, a mammal parasite responsible for pythiosis, shows a relatively recent divergence from *Pythium aphanidermatum* and *Pythium arrhenomanes*, both of which are plant pathogens [22]. There are many theories that explain how such drastic host shifts can occur in a small evolutionary timescale [23]. Particularly in oomycetes, large reservoirs of noncoding DNA material can readily evolve into small secreted proteins, known as effectors, facilitating new oomycete–host interactions [24]. Additionally, the readiness to take up genetic material through horizontal gene transfer from fungi and bacteria has been reported at multiple time points in the oomycete lineages [25,26,27]. However, the impact of host shifts on genomic selective pressures has not been extensively studied.

There is a high degree of convergent evolution between oomycetes and fungi [28]. Both share many of the niches mentioned, including pathogenic niches of animals and plants, as well as lifestyles, including saprotrophy, hemibiotrophy, and obligate biotrophy. Oomycetes and fungi have developed similar strategies to overcome the same challenges, including comparable filamentous and reproductive morphology, as well as akin infection strategies [29]. As mentioned above, convergence is probably promoted by genetic exchange, as the source of many oomycete genes with a role in host adaptation can be traced back to pathogenic fungi [30]. Because of the parallels between the adaptive strategies of these two eukaryotic phyla, we can infer underlying mechanistic principles in oomycetes on the basis of those further characterized in fungi.

How genome information relates to lifestyle and host adaptation is one of the big questions in ecology, and may be relevant to predict the appearance of new emerging diseases. Understanding the genome characteristics and selective pressures in organisms that have undergone host and niche shifts may offer insights into this question. In this study, we report the first whole-genome positive selection screening of the proteome of the oomycetes phylum, including 34 representative members and an outgroup of eight non-oomycete stramenopiles described in Appendix B. We compared the genes inferred as being under positive selection to the background annotated genes to identify enriched biological functions that may correlate to their adaption to different hosts and lifestyles. Additionally, we developed a method to predict plant pathogenic lifestyle with high accuracy from the genome of fungi and oomycetes, based on the presence or absence of key annotated functions.

## 2. Materials and Methods

### 2.1. Data Selection and Functional Annotation

We downloaded stramenopile genetic data from the NCBI and FungiDB databases setting as cutoff assemblies with reported gene annotation, resulting in a dataset of 42 total proteomes. We screened the genomes using BUSCO for high abundance of key orthologs in the stramenopile dataset as a form of quality control [31]. We performed functional annotation of the proteomes using InterProScan version 5.50–84.0 [32]. We validated proteins discussed in the manuscript through comparison with matches from the NCBI datatabase using BLAST [33]. We annotated the effectors in the stramenopile dataset by predicting the secretion signal using the tool SignalP 5.0b followed by an annotation with the model EffectorO [34,35]. We annotated the presence/absence of functional annotations from each genome with the Genome Properties database, performed the clustering with the Python library SciPy and visualized it with the package Seaborn [36,37]. We compared Unweighted Pair Group Method with Arithmetic Mean (UPGMA) clusterings of the genome properties and genome properties with added positive selection information to the phylogenetic tree using the Robison–Foulds metric based on clusters with the application TreeCmp [38,39].

### 2.2. Phylogeny Inference

We constructed the concatenated stramenopile tree using IQ-TREE 2 with automated partitioned model selection on inferred one-to-one orthogroups present in at least 25 of the taxa in the dataset [40]. We assessed full branch support in all nodes of the phylogenetic tree with 1,000 ultrafast bootstrap repetitions using the IQ-TREE 2 software and displayed it rooted on the outgroup of non-oomycetal stramenopiles.

### 2.3. Orthogroup Classification and Positive Selection Analyses

We developed a pipeline for whole genome positive selection analysis in Python using the Snakemake modular workflow framework [41]. It uses as input the coding nucleotide sequences as well as their corresponding predicted proteins from each proteome. The code and documentation are available at https://github.com/danielzmbp/wsgups (accessed on 24 June 2021). The steps of the pipeline include: grouping of sequences into ortholog families, their alignment with MAFFT, phylogenetic tree inference using FastTree, codon alignment using PAL2NAL, and finally two positive selection algorithms from the HYPHY package [42,43,44]. The first step, consisting of the classification of these proteomes into ortholog groups was performed with the software Proteinortho version 6, using the synteny parameter and the Diamond algorithm for homology search [45]. The first HYPHY algorithm used in the pipeline is FUBAR, a site-based program that scans the alignment for pervasive positive selection [46]. Families with at least one codon position under positive selection were subsequently analyzed on all branches with the aBSREL algorithm to relate selective pressures to specific lineages [47]. Taxa downstream of nodes with a corrected *p* value of less than 0.05 were considered under positive selection for this particular gene.

### 2.4. Enrichment Analyses

We used the Gene Ontology (GO) released in 1 February 2021 [48,49]. We performed GO enrichment using the Python package Goatools based on the InterPro database annotations [50,51]. The background dataset corresponded to the sum of all proteome annotations for the corresponding taxa and the study dataset to the genes found to be under selection. Terms that did not have representative sequences in all analyzed taxa were filtered out. We used as a significance cutoff the negative base 10 logarithm of Holm–Bonferroni corrected *p* values that were higher than 1.3 (*p* value < 0.05). Broad and non-informative GO terms like biological or cellular processes were not included in the enrichment tables.

### 2.5. Machine Learning Model

The multilayered deep learning models were constructed using the Tensorflow version 2.3 library with the Keras application programming interface [52]. The dataset consisted of 324 proteomes from fungi and oomycete plant pathogens as well as saprobes. We labeled each proteome as one of the four respective plant pathogenic classes based on the literature consensus: saprotroph, necrotroph, hemibiotroph and biotroph. For the genome properties model, we extracted the features of each genome and encoded them based on the presence or absence of all the identified pathways, which resulted in an array of 5024 binary features each. Removal of duplicated entries resulted in 319 unique samples. For the Carbohydrate-Active enZyme (CAZyme) model, we annotated the proteomes using the dbCAN2 database and encoded the absence or abundance of annotated CAZymes that were identified with the three implemented methods: HMMER, DIAMOND and Hotpep [53]. After removal of duplicates, 313 samples of 593 features were used for training and testing of the model. In both models, we performed a stratified split of the dataset into the training dataset, corresponding to 60% of the total, and the optimization and validation datasets, each corresponding to half of the remaining 40%. Hyperparameter optimization, namely of the learning rate, activating functions and dense layer units, was carried out using Keras Tuner and its implementation of the Hyperband algorithm [54,55].

## 3. Results

### 3.1. Proteome Annotation and Clustering

We downloaded the genomes of 34 oomycete species and eight non-oomycete stramenopiles from the NCBI and FungiDB databases and annotated their proteomes by the presence or absence of known functional signatures to gain insights into the divergence of the dataset (Figure 1) [56,57]. The UPGMA clustering based on the Euclidean distance along with midpoint rooting resulted in two main groups, one corresponding to the oomycetes and the other to the remaining stramenopiles. The main difference among them was the lack of photosynthesis-related annotations in the oomycetes, such as chlorophyll biosynthesis (Appendix A). In the oomycetes, we defined three clusters based on their distance (1–3 in Figure 1): obligate biotrophs, Saprolegniaceae, and a final one grouping most of the Perosporanaceae and Pythiaceae of the dataset. The obligate biotroph cluster consisted of the Albuginaceae and the downy mildews from the Peronosporaceae (*Bremia lactucae*, *Plasmopara halstedii*, *Peronospora effusa* and *Hyalopernospora arabidopsidis*). The most striking characteristic was an overall reduction in their metabolism, evident by the lack of many functional annotations in comparison with other oomycetes. This lack of core biosynthetic pathways, including vitamin and cofactor biosynthesis, makes them reliant on their host for growth and survival (Appendix A). The Saprolegniaceae group differed from other oomycetes mainly in the presence of steroid biosynthesis pathways (Appendix A). In the third cluster, we defined two subclusters, labeled as 3.1 and 3.2 in Figure 1. The first contained four of the *Pythium* and *Globisporangium* species of the dataset, and the second one included exclusively all *Phythophthora* in the dataset (except for *Phytophthora megakarya*). The *Pythium* and *Globisporangium* species in the dataset also had biosynthetic pathways that most other oomycetes lacked and that they often shared with the Saprolegniaceae, as a result most likely of their common facultative lifestyles. The hemibiotroph group, consisting of most of the *Phytophthora* species in the dataset, showed significant metabolic reduction, but not as extensive as in the obligate biotrophs [58].

These clusters and subclusters roughly reflected the lifestyles of the taxa in the dataset, mostly highlighted by the hemibiotrophs and obligate biotrophs. To a lesser extent, this was evident in the other two groups as most Saprolegniaceae in the dataset are facultative animal necrotrophs, and most *Pythium* and *Globisporangium* species facultative plant necrotrophs. Interestingly, *T. clavata*, the free-living organism in the dataset, clustered as an outgroup of the other phylogenetically close Saprolegniaceae, showing the greatest distance to its animal and plant-infecting neighbours. The most notable differences in the presence/absence of cellular pathways of this *T. clavata* assembly when compared to other Saprolegniaceae were the absence of the endopeptidase ClpXP complex and RuvB-like helicase I (Appendix A). However, there were some exceptions to this arrangement, with some taxa clustering with a different lifestyle or failing to cluster with their own lifestyle. For example, the clustering of the two plant infecting necrotrophs of the Saprolegniaceae follows the phylogeny of the *Aphanomyces* genus. Additionally, *Globisporangium splendes* appears as an outgroup of group 3.1 despite having a similar plant necrotrophic lifestyle and being phylogenetically closely related to other members in this clade (Appendix A). An explanation for this could be its long-read sequencing-based assembly. Long-read sequencing technology has been shown to produce much larger genomes in oomycetes when compared to the classical short-read sequencing, as hard-to-assemble repeat-rich regions are a common feature of their genomes [59]. However, other long-read assemblies in the dataset (*B. lactucae*, *Saprolegnia parasitica*, *Phytopythium vexans* and *Phytophthora fragariae*) show no apparent influence in the clustering (Figure 1).

### 3.2. Ortholog Group Classification

To infer positive selection from the stramenopile dataset of 42 genomes, we classified the proteomes into ortholog groups by taking sequence similarity and in addition gene order into account. We selected protein clusters that had at least five members from different taxa to obtain a good balance between a representative number of families and results that are statistically robust. This corresponded to 29,123 protein families, which cover about half (49.02%) of the total proteins in the dataset (Figure 2a). The orthogroups were mainly composed of one-to-one orthologs (78.70% of families); however, we detected a significant number of paralogs in some oomycetes, particularly for *Nothophytophthora* sp., as well as for *Phytophthora nicotianae*, *G. splendens* and *Phytophthora parasitica* (Figure 2b). This might be related to the reported whole genome duplications in *Phytopthora* species [60], as well as the recent hybridization event that gave rise to *Nothophytophthora* [61]. Additionally, the diatom *Fistulifera solaris*’s large presence of gene duplications highlights its recent whole genome duplication [62]. The larger presence of duplicates in *P. insidiosum* and *G. splendes* in comparison to their peers may originate from their long-read sequencing-based assembly, which is better able to resolve gene duplications.

The most abundant orthogroups had between five and nine members (Figure 2c). Orthogroups corresponding to all taxa were a minority. Instead, most orthogroups were present in closely related five- to ten-member clades. When looking at the number of genes not assigned to orthogroups in the oomycetes, the *Phytophthora* genus had the highest count (Figure 2a). This may be related to the large arsenal of unique effectors that lack conserved domains or homologs outside of their own species and play a large role in host adaptation. *Aphanomyces astaci* also had a high amount of genes outside of the orthogroups, most likely because of the recent expansions in its genome [63]. In summary, this highlights a patchy ortholog distribution in the dataset, with most protein families conserved only in phylogenetically close members of clades (Figure 2c). Despite this, a significant pool of ortholog protein families representative of the stramenopile genomes in the dataset could be inferred from the analysis as further discussed below.

### 3.3. Positive Selection Analyses

Positive selection screening for orthologous groups was performed by using first a site-specific codon model to detect families under selection. This was followed by a branch-site-specific codon model to detect the taxa experiencing positive selection on those genes. The number of genes under selection varied for the different phylogenetic clades. Members of the Saprolegniaceae and Pythiaceae, together with the necrotrophic *Globisporangium* had a higher count and therefore more genes under selection in orthogroups (mean = 1222, std = 152) than the remaining Peronosporaceae and the Albuginaceae (mean= 577, std = 245) (Appendix A). A special case was the hybrid *Nothophytophthora* sp., which had a comparable amount of positively selected genes to Pythiaceae and Saprolegniaceae, however, composed in great part by duplicated genes after speciation, 44.45% of the total (orange bar). When comparing necrotrophs, hemibiotrophs, and obligate biotrophs within the Peronosporaceae family (mean = 1344, 663, and 269, respectively), the trend was that of a decrease in the number of genes under positive selection with the increase in biotrophic potential (Figure 3).

To infer potential biases in our analyses, we tested for a correlation between the number of genes under positive selection and the amount of proteins classified into orthogroups for each taxa (Pearson’s correlation, r = 0.50, *p* value < 0.01). A correlation of 0.5 suggested that there may be a larger number of positives because of more extensive testing in the oomycete species, as they have on average more members in the ortholog dataset. This bias is more evident in the non-oomycetes (Pearson’s correlation, r = 0.52, *p* value = 0.18) than when considering just the oomycetes (Pearson’s correlation, r = 0.15, *p* value = 0.39). As the proteomes of the non-oomycetes are overall smaller compared to oomycetes (Appendix A), we hypothesize that less extensive testing renders them more prone to this bias.

Out of the 32,661 detected genes under positive selection, 21,247 were successfully annotated with at least a GO term (65%). We performed GO enrichment on the four main oomycete lifestyles in the stramenopile dataset. The results are discussed below. As a control for the reliability of the pipeline, we performed the same analyses in a subset of 26 plant pathogens from a dataset of 65 basidiomycete fungi (Appendix A). Highly enriched terms included processes known to be associated with virulence in such pathogenic fungi, like fatty acid and certain amino acid biosynthesis, ion transport, and protein targeting and transport (Appendix A) [64,65,66].

In summary, we could identify signatures of positive selection in 4.14% of all genes analyzed in the stramenopile dataset. A significant number could be functionally annotated and potential functions assigned.

### 3.4. Enriched Biological Functions under Selection

To gauge the selective pressures for adaptation to a parasitic lifestyle in the oomycetes, we explored the enriched GO terms that were pervasive in all oomycetes (Figure 4a). Highly enriched term categories related to response to stress, signal transduction, transmembrane transport, protein modification processes (phosphorylation, in particular), and localization, as well as numerous carbohydrate, lipid, nitrogen, and sulfur metabolism-related terms. Within the metabolism, abundant terms relating to biosynthesis are present. In the cellular compartment GO category, highly enriched terms include protein-containing complexes (for which transferase complexes show the larger significance), nucleus, intracellular organelles (for which ribosome shows the largest significance), and membranes (Figure 4b).

Additionally, we performed similar enrichments on the oomycete groups as defined by their lifestyle. We found the largest amount of unique GO terms to belong to the plant and animal necrotrophs (36 and 21, respectively). In the plant necrotrophs, these included terms related to ion transport, carbohydrate biosynthesis, protein modification, and gene expression regulation. In the animal necrotrophs, unique terms had to do with vitamin biosynthesis, cilium movement, and protein localization. There were three unique terms in the hemibiotrophs related to response against stress and transmembrane transport while no unique terms were identified in the obligate biotrophs. We observed the largest overlap between animal and plant facultative necrotroph groups (59 common terms). These terms related to cell communication, glycolysis, organelle assembly, protein import, regulation of response to stimulus, translation, and numerous and diverse metabolic processes. This was followed by a smaller overlap of enriched functions in all four lifestyle groups, amounting to 33 terms (Figure 4c). The significant terms for each lifestyle are all listed in Appendix A.

We also studied the enrichment of biological functions in the expanded gene families of the dataset independently of whether the genes were under positive selection. In general, we found that it reflected positive selection enrichment; however, the terms were highly variable when comparing different species (Appendix A). In the obligate biotrophs, these related to phospholipid metabolism, cell wall biosynthesis, protein modification, biological regulation, and transmembrane transport. In the hemibiotrophs, they related to lipid metabolism, signaling, protein modification, and again to biological regulation, and transmembrane transport. Finally, in the plant necrotrophs, they related to DNA integration and localization.

### 3.5. Lifestyle Prediction

We visualized in a heatmap all functional annotations with added information of positive selection by performing the same clustering as we did for the genome properties (Figure 5). We find that adding the positive selection data improves the clustering by lifestyle, particularly of the plant necrotrophs in the Pythiaceae and *Globisporangium*, which now form a single cluster that is closer to the other facultative necrotrophs of the dataset, the Saprolegniales, than to the obligate biotroph and hemibiotroph oomycetes in the dataset. Using the Robison–Foulds metric for clusters, we observed that there is a higher congruence between the phylogenetic tree and the genome properties clustering than to the positive selection one (Table 1).

Although positive selection information improved lifestyle clustering, we argue that it is impractical to implement as a prediction method because of its computationally demanding calculation and poor reproducibility when using different backgrounds for positive selection analyses. Therefore, we constructed a model to predict lifestyle in plant pathogenic fungi and oomycetes based on the genome properties alone. We assembled a dataset comprising 324 genomes from 115 plant pathogenic and saprotrophic fungal and oomycetal species (Appendix A). We used the annoted genome properties as features to build a deep neural network classifier with four output classes corresponding to their lifestyle consensus in the literature: saprotroph, necrotroph, hemibiotroph and biotroph. We found a high accuracy on the validation dataset for the optimized model (loss = 0.11, accuracy = 0.95), failing to predict two genomes in the hemibiotrophs and one in the biotrophs of the validation dataset (Figure 6). For comparison with published models, we additionally constructed a predictor based on Carbohydrate-Active enZymes (CAZymes), which also resulted in a high accuracy for the random validation dataset and performed better in the prediction of hemibiotrophs (loss = 0.14, accuracy = 0.97). Both models and the steps to reproduce them together with the entire dataset can be found at https://github.com/danielzmbp/lspred (accessed on 24 June 2021).

## 4. Discussion

### 4.1. Functional Genome Annotations Largely Correlate with Lifestyle

Convergence of the presence/absence of key functional annotations in species that do not share the same phylogenetic history but have similar lifestyles has been shown before for different sets of organisms [67,68]. Distant species with the same lifestyle require similar functional biological processes, which results in similar selective pressures that analogously shape their genome, often leading to convergent evolution. Comparable to the study by Rodenburg et al. (2020) [69], we have shown the tight clustering of some oomycete groups with a similar lifestyle, most strikingly for the obligate biotrophs and hemibiotrophs. Conversely, we find a few exceptions in our dataset, such as the hemibiotroph *P. megakarya* and the necrotroph *G. splendens*, which do not clearly cluster with any of the other oomycetes. We hypothesize this may be partly due to the quality of their gene annotation. Both have a significantly lower number of complete key orthologs than the reference stramenopile BUSCO database as compared to other *Phytophthora* and *Globisporangium* species in the dataset (Table A1).

### 4.2. Generalists Have More Genes under Positive Selection

A higher number of genes under selection was found for the more generalist families of Saprolegniaceae, Pythiaceae, and necrotrophic Peronosporaceae, including the *Globisporangium* and *Phytopythium* clades, when compared to the higher number of specialists remaining in Peronosporaceae and Albuginaceae (Mann–Whitney test, *p* < 0.01). Within the Peronosporaceae, hemibiotrophs have a lower number of genes under selection than the facultative necrotrophs, and obligate biotrophs have in turn a lower number than hemibiotrophs (ANOVA one-tailed test, *p*< 0.01) (Appendix A). Thus, the number of genes under selection is inversely correlated to the biotrophic potential. With biotrophic potential, we refer to the capability of the pathogen to survive exclusively on a living host, such that no obligate biotroph can be cultured in vitro, while for some hemibiotrophs this is the case. On the opposite side of the spectrum, facultative plant necrotrophs thrive as saprotrophs without the need for a host. This correlation cannot be explained alone by the different sizes of the proteomes in the dataset or by their phylogenetic closeness (Appendix A). However, we hypothesize that both of these factors confound our results to a large extent. Smaller proteomes in the dataset, as is the case of the non-oomycetes, show a larger correlation of their size to the number of genes under positive selection. The phylogeny influence is highlighted by the similar number of genes under positive selection of taxa within the same genus as shown in Appendix A.

While all hemibiotrophs and biotrophs are obligate plant parasites, the necrotrophs in the Peronosporacea, Pythiaceae and Saprolegniaceae families show adaptation to a variety of lifestyles. They are facultative parasites of either animals, plants, or other fungi and oomycetes. Facultative parasites can live as saprotrophs on decaying matter but also as opportunistic necrotrophs on a suitable host [70]. The higher number of potential niches they are able to successfully occupy may drive a larger number of genes to be under positive diversifying selection. Additionally, when compared to the obligate biotrophs and hemibiotrophs, which are highly adapted to infect a particular species, e.g., lettuce for *B. lactucae* and soybean for *Phytophthora sojae*, most of the necrotrophs are able to infect a wide range of hosts. For instance, *A. astaci* is capable of infecting up to twelve genera of crayfish and is known for its ease of host jumping [71]. Having a higher number of genes under positive selection could be therefore correlated with this higher host flexibility.

### 4.3. Selective Pressures in the Ooomycetes Help Explain Host Adaptation

Biological functions under selection for all oomycetes in the stramenopile dataset, shown in Figure 4, give insight into which of these are important for the diversification in this clade. Many biosynthetic functions, particularly related to carbohydrates, are found to be enriched. Different cell wall carbohydrate composition has been found for different clades in the oomycetes, likely associated to different lifestyles [72]. Lipid metabolism, known also to be important for host adaptation in plant pathogenic fungi and oomycetes, is also enriched [73]. Transport-related proteins, and in particular cation transport, are also prominently enriched in these terms. As an example, the role of the expanded calcium transporter genes in the oomycetes has been extensively studied in the context of host interaction [74]. Overall, many of these terms allude to important virulence factors known for the oomycetes: transmembrane transport, effector protein processing and secretion, cell wall synthesis and remodeling, and lipid localization [75].

#### 4.3.1. Selective Pressures Relate to Lifestyles in Oomycetes

The enriched terms common to the Albuginaceae and downy mildews greatly relate to known virulence factors for these plant pathogens, including carbohydrate metabolism, protein modification, transport, negative regulation of gene expression, and response to stimuli (Appendix A). This suggests that these biological functions are under selection and played a big role in the adaptation of oomycetes to an obligate biotrophic lifestyle. Some of these, particularly carbohydrate metabolism, transport, and protein modification, are common to the other plant pathogens in the hemibiotrophs and plant necrotrophs (Appendix A), highlighting a broader mechanism of adaptation to a plant-parasitic lifestyle.

One of the most often found terms and among the most enriched in both the obligate biotrophs and the hemibiotrophs of the dataset corresponds to regulation of biosynthetic and metabolic processes, and particularly negative regulation. This may underscore the fitness advantage for rapid growth during the hyphal stage and its need for activation or deactivation according to the circumstances. When the hyphal stage takes place after colonization, the salvaging and biosynthesis of carbohydrates, nucleic acids, and lipids with the resources obtained from the plant host is key for a successful infection. Beta-glucan, for example, is an important component of the oomycete’s cell wall and is also an elicitor of the plant immune response [76]. Its biosynthesis features prominently in the enriched terms for the hemibiotrophs. Following the hyphal stage and massive host colonization, there is a reproductive, mainly sporulating stage which needs to be tightly regulated. The latent period between infection and sporulation has been shown recently to correlate to different pathogenic lifestyles in leaf pathogens [77]. A widely diversified family of proteins in the oomycetes that take part in transcriptional activation and help regulate sporulation are Myb transcription factors [78,79]. The ortholog family of the gene *Myb3R7*, which was shown in *P. infestans* to be upregulated during sporulation, shows high rates of positive selection for the *Aphanomyces* and *Pythium* clades. Another gene associated to sporulation, classified as *CDC5*, was also found to be under selection in the oomycetes of the stramenopile dataset.

Secretion of small effector proteins, as in other fungal filamentous pathogens, is key for host adaptation in plant pathogenic oomycetes [80]. Many unique effector proteins have been characterized in the oomycetes that contribute to virulence by modulating the immune response of the plant [81]. Therefore, this dependence on the secretion machinery of the cell for successful infection and thus survival has led to high selective pressures on their genome. We observed significant enrichment of the effectors in the positively selected terms in all oomycetes of the dataset (hypergeometric test, *p* < 0.01). When looking at the enrichment per species; the majority of the *Phytophthora* and plant necrotrophs, which significantly depend on effector proteins for host infection, were also enriched (Figure 7a). The obligate biotrophs, which also depend greatly on secreted effectors, do not show enrichment in our analysis. This may be due to the lack of orthologs on host specific effectors and thus not them not being analyzed in the positive selection screen. There is a small correlation between the number of positively selected genes compared to those predicted to be effectors (Pearson’s correlation, r = 0.22, *p* = 0.22), so these results may be slightly skewed due to testing bias (Figure 7b). In the GO enrichment of all oomycetes, there were several processes directly related to protein secretion under selection, including protein modification. Other secretion-related terms, although more general, also showed enrichment, including those relating to microtubule-based processes in the obligate biotrophs, and transmembrane transport in the hemibiotrophs.

Another interesting term indirectly related to effector proteins is sulfur amino acid biosynthesis. This term is highly enriched in the hemibiotrophs and the necrotrophs of the dataset. This may be associated with the abundance of cysteine-rich proteins in the effector arsenal of the plant pathogens with a necrotroph phase [82]. The disulfide bonds that link cysteine residues help maintain the structural integrity of the proteins released into the extracellular space called apoplast, a hostile environment that is slightly acidic and rich in plant proteases [83]. In general, elevated amino acid biosynthesis has been shown to be an important factor during infection of plant hemibiotrophs such as *P. infestans* [84]. We found enzymes related to amino acid biosynthesis, namely methionine synthase and ketol-acid reductoisomerase, to be under pervasive positive selective pressure in most oomycetes in the stramenopile dataset. These were previously found to be upregulated during early infection and the switch to necrotrophic phase in *P. infestans* [84].

When looking exclusively at the necrotroph groups, many terms in the plant pathogens overlap with the animal pathogens, most likely relating to their facultative saprobe lifestyle. These include glycolysis, generation of energy, cell communication, as well as amino acid, tetrapyrrole, and amide biosynthetic processes. The latter group is most likely enriched as a result of their autotrophic and more developed secondary metabolism compared to that of other oomycetes, which makes them suited to a free-living lifestyle [85]. The highest enriched term under positive selection for both groups corresponds to the inosine salvage pathway, in particular the enzyme inosine 5’-monophosphate-specific 5’-nucleotidase. Nucleoside accumulation in the plant apoplast has been shown to increase susceptibility to the plant fungal necrotroph *Botrytis cinerea* [86]. A similar case in oomycete necrotrophs could have led to high selective pressure on this enzyme. Equally interesting is the term DNA ligation involved in DNA repair, which may be related to the defense against oxidative stress that is key to the immune response in plants and animals against such pathogens [87].

#### 4.3.2. Biosynthetic Repertoire Is Important for Lifestyle Adaptation

As shown in Figure 1, the biosynthetic repertoire of each taxa plays a big role in defining the lifestyle of the organisms in the stramenopile dataset. Particularly interesting in oomycetes is the evolutionary history of sterol *de novo* biosynthesis. It is present in Saprolegniales and absent in other oomycete lineages due to their inability to synthesize oxidosqualene [88,89]. The squalene synthase shows hints of positive selection in *Aphanomyces* (Appendix A). Furthermore, positive selection is pervasive in the enzymes that take part in sterol biosynthesis in the stramenopile dataset.

Vitamin biosynthesis as well plays a big role in the evolution of pathogen adaptation to its host. Vitamins are expensive to produce and often require dedicated pathways. Heterotrophs that have adapted to obligate biotrophic lifestyles, such as *Albugo* and the downy mildews, circumvent this by losing their biosynthetic capabilities and developing ways of utilizing host vitamin supply, also known as auxotrophy [90]. Meanwhile, those that live without a host at any point in their lifecycle must maintain these pathways under strong purifying selection. In our dataset we found signatures of positive selection in several enzymes relating to tedrahydrofolate (THF) salvage and biosynthesis, namely dihydrofolate synthase and phosphoribosylglycinamide formyltransferase (Appendix A). As THF is a derivative of Vitamin B9 or folic acid, it is crucial for the synthesis of several amino acids such as serine and methionine as well as for purines and thiamine [91]. It is therefore likely that oomycetes that are not able to obtain THF from a living host have strong selection to maintain THF metabolism in order to ensure their own amino acid biosynthesis.

Molybdopterin cofactor is important for the production of certain detoxification enzymes [92]. In oomycete obligate biotrophs, molybdopterin-related biosynthetic pathways have been lost independently several times in the oomycetes’ lineage due to host adaptation [15]. Molybdopterin metabolism was found under high selective pressure in the facultative necrotrophs and autotrophs of the stramenopile dataset, including *Saprolegniaceae* and *Pythiaceae* families, and *Phytophthora* genus (Appendix A). The biosynthesis of molybdopterin cofactor also features as an enriched GO term in the plant necrotrophs (Appendix A).

Proteins relating to the glycolysis pathway and amino acid biosynthesis have a special evolutionary history in the oomycetes [93]. Many of these enzymes originated from horizontal gene transfer from plants or bacteria. This might explain their high rate of positive selection, which is usually the case for genes recently acquired by horizontal transfer, as they need to be adapted to the new host. In the glycolysis pathway, we detected signatures of positive selection for most oomycetes in the stramenopile dataset, particularly in the enzymes glyceraldehyde-3-phosphate dehydrogenase and fructose-bisphosphate aldolase (Appendix A).

#### 4.3.3. Protein Family Enrichment Reflects Lifestyle Selective Pressures

The large overrepresentation of paralogs as positively selected genes is evident in many of the taxa (Figure 3). After a gene duplication event occurs, there is usually an increase in the selective pressure on one of the copies that maintains the function. Meanwhile, in the other one, these constraints are relaxed, freeing it for potential divergent evolution [94]. Interestingly, many of the enriched functions in the paralogs correlated with terms under positive selection for their specific lifestyle (Appendix A). In the *Phytophthora* lineages these include biological regulation, glycolipid biosynthesis, and transmembrane transport. In *Albugo* and other obligate biotrophs, protein modification, carbohydrate metabolism, biological regulation, and glutamine metabolism.

### 4.4. A Model Based on Genome Properties Accurately Predicts Lifestyle

The genome convergence of phylogenetically diverse fungi and oomycetes allowed us to create a model that can predict plant pathogenic lifestyle based on annotations from both eukaryotes. Assessment of lifestyle from genomic properties in plant pathogens has been traditionally performed by characterizing cell wall-degrading enzyme annotations [95]. To our knowledge, there is only one other published model that attempts to predict lifestyle from genomic features [96]. This model predicts trophic categories based on principal component analysis of CAZyme annotations. We find that our model, which in contrast is based on entire genome annotations, allows for better overall accuracy. However, its better performance may be partly related to its four output lifestyles compared to the seven classes in the model from Hane et al. (2020), since when comparing the genome properties model to a four-class model trained using similar parameters on CAZymes, the prediction is comparably precise (Figure 6). An advantage of the genome properties model is that having trained it on a larger number of features per sample allows for a more accurate prediction of incompletely annotated specimens that may result from environmental sampling. Given the availability of increasing proteomic and transcriptomic data of unknown fungal and oomycetal origin, such prediction tools will become crucial to identify the pathogenic potential of facultative and obligate parasites.

## 5. Conclusions

The presence/absence of metabolism-related genes is known to converge for phylogenetically distant organisms that follow the same lifestyle [69,97]. Here, we report a similar case for our dataset of stramenopiles. We developed a pipeline for seamless throughput analysis of positive selective pressures using genome data as input and employed it to show that patterns of selective pressure also converge on hosts that cannot be explained by phylogeny alone. Clustering the taxa by their positive selection predictions, we showed improvement in the prediction of their lifestyle. Furthermore, we identified a number of biological functions that are commonly found under selection for all oomycetes independently of lifestyle. We explored and discussed lifestyle-specific adaptive genes that corresponded to biological regulation, transport, protein modification and metabolite biosynthesis. Finally, we described a model based on genome properties that is able to accurately predict the plant pathogenic lifestyle of filamentous fungi and oomycetes.

Overall, we believe our results highlight to date unexplored genes that could lead to further understanding of lifestyle evolution in oomycetes, and more broadly in filamentous pathogens. Experimental exploration on the impact these adaptations may have in the function and behaviour of the encoded proteins could improve prevention and prophylaxis against new emerging pandemic threats. Particularly, the enzymes reported in this manuscript, due to their likely adaptation as a consequence of a shift in host or lifestyle, may present interesting new targets which could be further explored for disease control.

## Figures and Tables

**Figure 1 pathogens-10-00807-f001:**
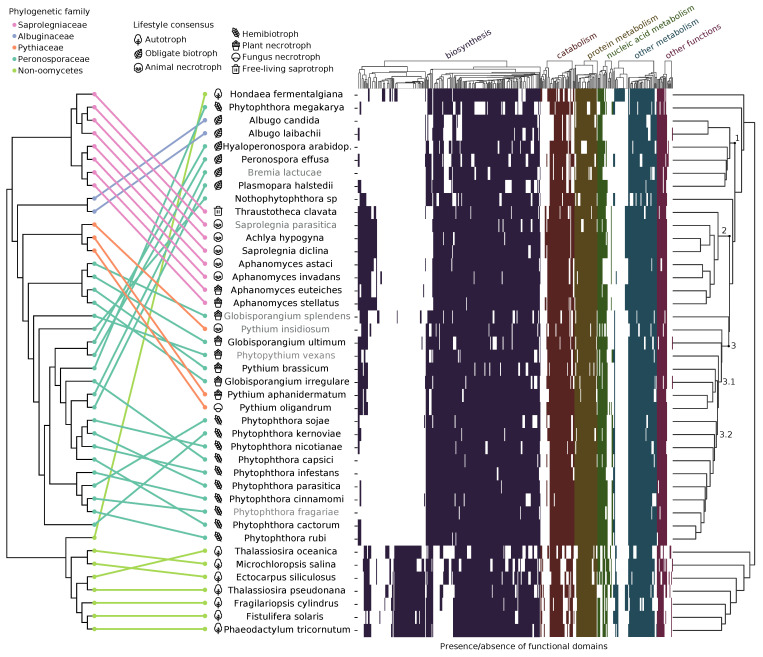
Presence/absence of functional attributes in the genomes of the stramenopile dataset correlated with phylogeny. Equal distance cladogram constructed from conserved families inferred by the maximum likelihood on the left and clustering by Unweighted Pair Group Method with Arithmetic Mean (UPGMA) of genome properties of the dataset on the right. In the equal distance phylogenetic tree, colored lines match phylogeny to the clustered taxa with annotated lifestyles. All nodes in the tree have a 100-bootstrap support. In the heatmap, different colors represent the presence or absence of particular functional groups belonging to the specified categories. Names in grey represent long-read sequencing assemblies in the dataset for that species.

**Figure 2 pathogens-10-00807-f002:**
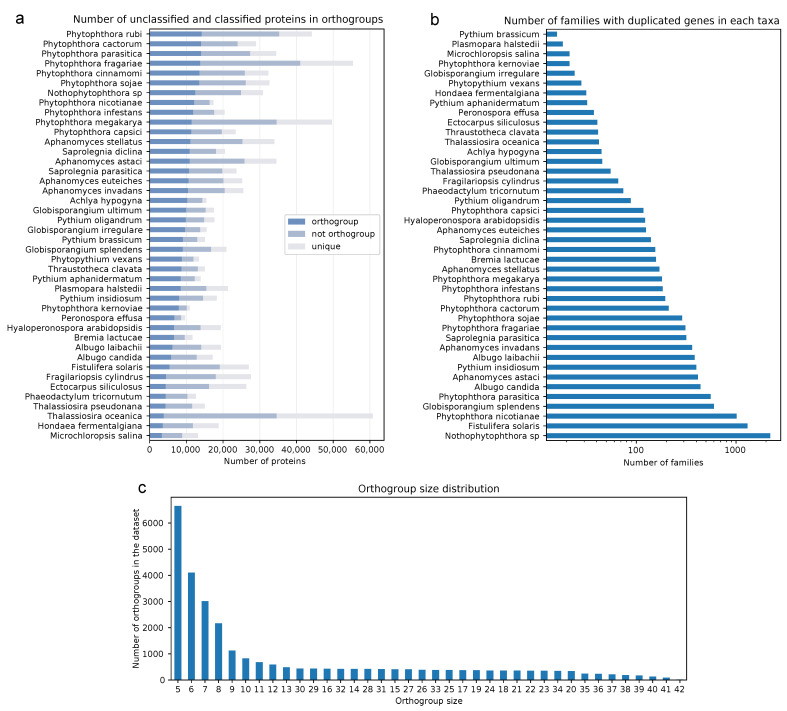
Orthogroup classification in the stramenopile dataset. (**a**) Protein-encoding genes from the dataset grouped into ortholog families. Number of genes classified into orthogroups (protein families of five or more members), not classified into orthogroups (protein families of less than five members), or unique (not in a protein family) are displayed per taxa. (**b**) Duplicates in protein families of the dataset. Number of ortholog families with five or more members from different taxa that contain paralogs (two or more genes from the same taxa). (**c**) Distribution of protein family size in the dataset. Number of families with the same member size are represented as a histogram.

**Figure 3 pathogens-10-00807-f003:**
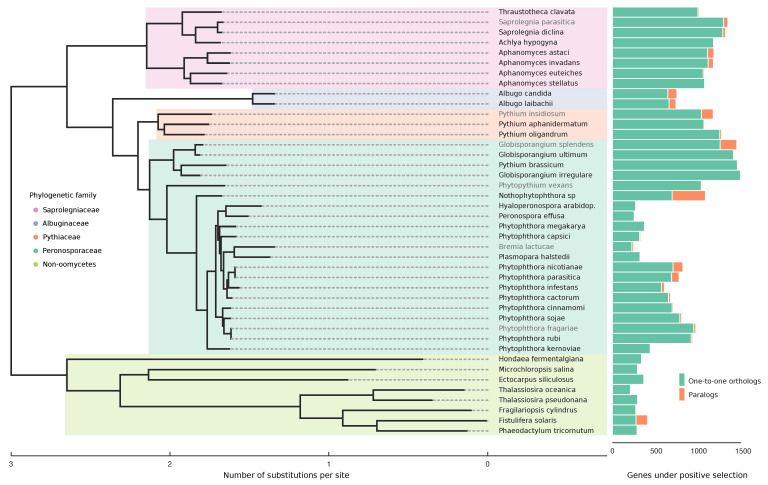
Number of genes under positive selection in the stramenopile dataset. The Maximum likelihood supertree with 100-bootstrap support in all nodes was constructed from inferred protein families in the stramenopile dataset that are conserved in at least 25 taxa, corresponding to 3013 families of orthologs. A number of positively selected genes are represented as bars. One-to-one orthologs are in green, duplicated genes inside the same family under positive selection in orange. Names in grey represent long-read sequencing assemblies in the dataset for that species.

**Figure 4 pathogens-10-00807-f004:**
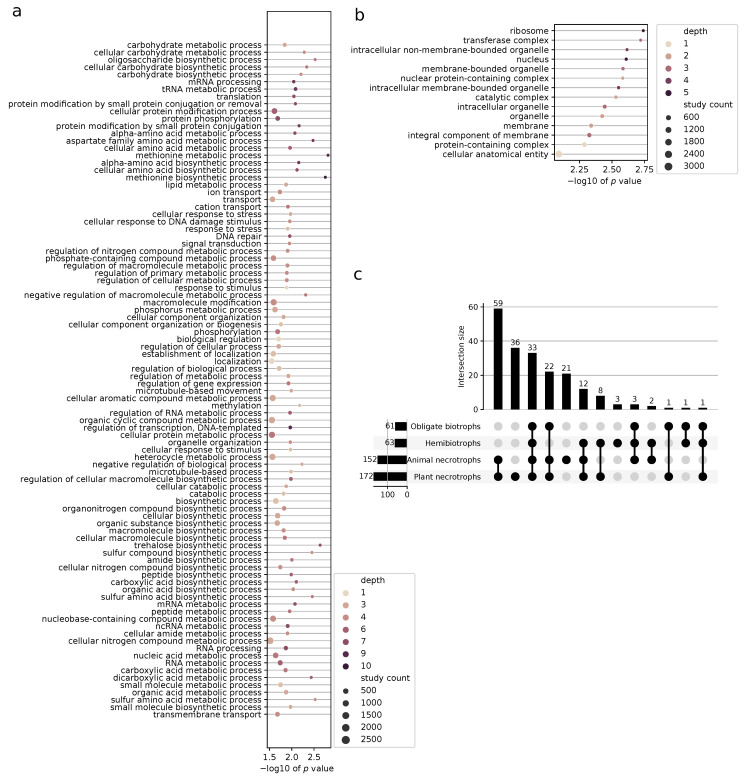
Functional and location enrichment of positively selected genes in the oomycetes. Significantly enriched (**a**) biological processes and (**b**) cellular compartments in all oomycetes that show signals of positive selection in the stramenopile dataset. Included are gene ontology (GO) terms with a corrected negative base 10 logarithm of the *p* value higher than 1.5 ordered by category using the GO slim database. The color represents the GO depth. GO depth is a measure of the number of parent nodes in the GO tree. That is, the more specific the GO term, the higher its depth. The size of the dots corresponds to the total number of proteins under selection in the stramenopile dataset that belong to said term. (**c**) Upset plot showing number of overlapping biological functions under selection in the oomycetes. The four groups correspond to the major lifestyles in the oomycetes of the stramenopile dataset.

**Figure 5 pathogens-10-00807-f005:**
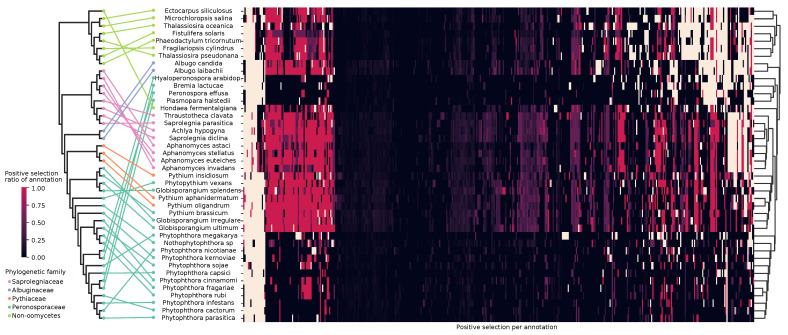
Heatmap of positive selection ratio of functional annotations in the stramenopile dataset. The color gradient from black to red in the heatmap represents the ratio of genes with a particular functional annotation that are under selection. Cream-colored cells represent the absence of the annotation for that species. The phylogenetic equal distance cladogram is represented on the left while UPGMA-based clustering of the distance between the taxa is represented on the right. Colored lines representing the phylogenetic family connect both clusterings.

**Figure 6 pathogens-10-00807-f006:**
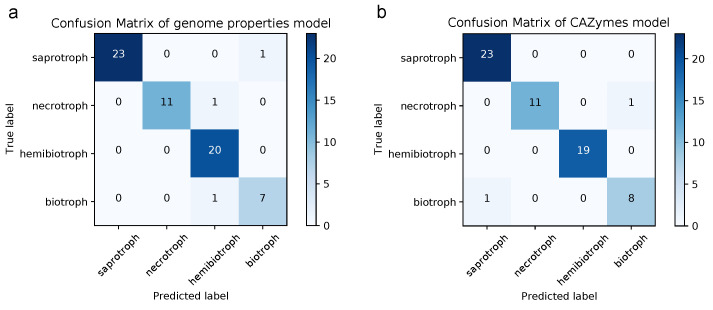
Confusion matrix of lifestyle predictor models. Prediction results in the random validation sets for the constructed models based on (**a**) genome properties and (**b**) Carbohydrate-Active enZymes (CAZymes), corresponding to 64 and 63 annotated proteomes, respectively (20% of the total dataset). True values are represented on the *x*-axis and predicted values on the *y*-axis.

**Figure 7 pathogens-10-00807-f007:**
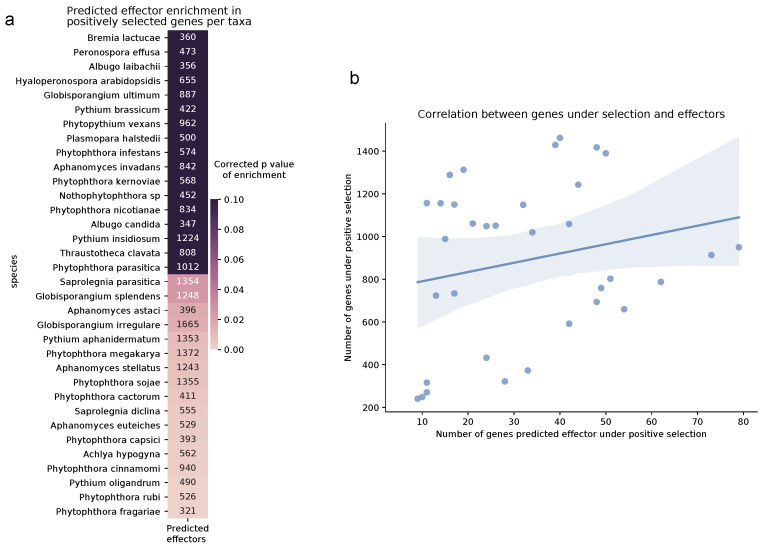
Effector proteins in the oomycetes of the stramenopile dataset. (**a**) Enrichment of genes coding for effector proteins under positive selection in oomycetes. The color gradient represents significant *p* values from hypergeometric tests per taxa corrected for multiple testing using Bonferroni. A lighter shade represents a more significant enrichment. The numbers within the cells represent the total effectors per proteome in the stramenopile dataset that were analyzed for positive selection. (**b**) Correlation between genes under selection and effectors in the oomycetes. Pearson correlation is represented as a straight line and the confidence interval is represented as a lighter shade (r = 0.22, *p* value = 0.22).

**Table 1 pathogens-10-00807-t001:** Distance comparisons in the clusterings of the stramenopile dataset. Phylogenetic and genome properties clustering is shown in Figure 1 and positive selection clustering in Figure 5.

Clustering 1	Clustering 2	Robison–Foulds Distance Metric
Phylogenetic	Genome properties	28
Phylogenetic	Positive selection	30
Genome properties	Positive selection	24

## Data Availability

The stramenopile data presented in this study are openly available in Zenodo at https://doi.org/10.5281/zenodo.4725040 (accessed on June 2021).

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
