# Peer review of "Predicting Lifestyle from Positive Selection Data and Genome Properties in Oomycetes"

_pathogens, 2021, doi:10.3390/pathogens10070807_

Round 1

Reviewer 1 Report

In this paper, the authors develop a comparative genomics approach to predict lifestyle and virulence among oomycetes. This method is based on correlations between genome properties and possible evolutionary trends that are deduced from genome annotations.

At first sight, the approach is appealing and would undoubtedly contribute to better understanding adaptive trends on oomycetes, especially on plant pathogens. Yet, this study has serious flaws.

A major concern is that the paper relies on the comparison of 42 proteomes that either derive from long-read or short-read genome sequences. The resulting assemblies may greatly differ according to the sequencing methodology, with high impact on the extent and complexity of global predicted proteomes. This is particularly important in the case of oomycete genomes which are characterized by a high proportion of repetitive sequences and expansion of protein-encoding gene families. Duplicated sequences and repeats may be collapsed in short-read assemblies. Poorly resolved sequences include gene-sparse regions that host transposable elements and genes relevant to parasitism (effectors). Recent re-sequencing projects led to largely bigger (>50%) genome sizes, accompanied with increased BUSCO scores, revealing duplication rates higher than expected. So, a global comparison of proteomes derived from various sequencing projects merits more caution.

Another flaw is that this study also relies on comparison of cellular pathways that are solely deduced from enrichment analyses of GO terms. Here again, the peculiarities of oomycetes may in some cases contradict automatic functional annotation. The authors should at least initiate the validation of some hypotheses presented in this paper by exploration of experimental data.

Author Response

We greatly appreciate your comments and constructive feedback. The responses to your two points are as follows:

Point 1: A major concern is that the paper relies on the comparison of 42 proteomes that either derive from long-read or short-read genome sequences. The resulting assemblies may greatly differ according to the sequencing methodology, with high impact on the extent and complexity of global predicted proteomes. This is particularly important in the case of oomycete genomes which are characterized by a high proportion of repetitive sequences and expansion of protein-encoding gene families. Duplicated sequences and repeats may be collapsed in short-read assemblies. Poorly resolved sequences include gene-sparse regions that host transposable elements and genes relevant to parasitism (effectors). Recent re-sequencing projects led to largely bigger (>50%) genome sizes, accompanied with increased BUSCO scores, revealing duplication rates higher than expected. So, a global comparison of proteomes derived from various sequencing projects merits more caution.

Response 1: We have explored the BUSCO genome scores of oomycete assemblies from NCBI and we can confirm the pattern here mentioned of a larger number of duplicated BUSCOs for long-read sequencing-based assemblies (figure 1 of the attached pdf). However, we argue this is a small effect unlikely to greatly affect our results and conclusions. When we look at the BUSCO scores per species for all assemblies available in NCBI for oomycetes in our dataset we see no significant difference in the number of single and duplicated BUSCOs. An exception is Phytophthora infestans which shows around 10 more duplicated BUSCOs in comparison to short-read sequencing assemblies (figure 2 of the attached pdf).

We have added the sequencing read length annotation to the main figures of the manuscript and added a small discussion on sequencing read length. However, apart from Globisporangium splendes in figure one, which contrary to our expectations does not cluster with similar species in their lifestyle or phylogeny, we believe that using both sequencing methods in the dataset had no major impact on the results and conclusions. Additionally, in this study we only look at the more conserved proteins, since these are the ones that we could functionally annotate with confidence and find the orthologs for. Most host-specific effectors, which are more likely to be misannotated in the shorter read sequencing, were not used in the analyses.

Point 2: Another flaw is that this study also relies on comparison of cellular pathways that are solely deduced from enrichment analyses of GO terms. Here again, the peculiarities of oomycetes may in some cases contradict automatic functional annotation. The authors should at least initiate the validation of some hypotheses presented in this paper by exploration of experimental data.

Response 2: We have validated the automatic GO term annotations by sequence similarity for some of the more interesting candidates. We have also expanded on several points in the discussion where we link genes found under positive selection in our analysis with experimental data from other publications.

Reviewer 2 Report

In Oomycetes, different species among different families have similar lifestyles, showing convergent evolution. Although this phenomenon has been widely observed, it's not clearly explained before. In this manuscript, converging evolution of oomycetes was revealed by comparative genomics approach. Processes are driven by adaptive evolution, which is related to certain functions, in particular carbohydrate metabolism, transport, and signaling for host and environmental adaptation in oomycetes. On that basis, a model that predicts plant pathogenic lifestyles was developed, providing new vision for understanding 15 host/lifestyle shifts and their impact on the genome.

I only have two minor comments/remarks for a revision:

  1. In Figure 1, the equal-distance phylogenetic tree with colored lines matching phylogeny to the clustered taxa with annotated lifestyles was constructed. However, there is no information on construction of phylogenetic tree and lifestyles annotated according to In addition, no support values are marked on tree. If possible, the contents should be added as supplemental file.

The figure 16 is important for understanding positive selection ratio of functional annotations. So phylogenetic tree with the clustered taxa should be added to the left side of figure 16.  

Author Response

We greatly appreciate your comments and constructive feedback. The responses to your two remarks are as follows:

Point 1: In Figure 1, the equal-distance phylogenetic tree with colored lines matching phylogeny to the clustered taxa with annotated lifestyles was constructed. However, there is no information on construction of phylogenetic tree and lifestyles annotated according to In addition, no support values are marked on tree. If possible, the contents should be added as supplemental file.

Response 1: We have added phylogenetic tree construction information to the legend in Figure 1. We have also added the sources for the lifestyle annotation on the stramenopile dataset to Table 1 as suggested.

Point 2: The figure 16 is important for understanding positive selection ratio of functional annotations. So phylogenetic tree with the clustered taxa should be added to the left side of figure 16.

Response 2: We agree that adding the phylogenetic tree to this figure would be an improvement. Therefore, we have added the annotated phylogenetic tree to the left of Figure 16 and matched it to the positive selection clustering on the right.

Reviewer 3 Report

This is a very interesting and overall well-written manuscript. I have only a few comments for authors to consider.

  1. The title is a bit misleading. While the paper demonstrated the predictive power of the positive selection data and genome properties for the four lifestyles of oomycetes, I didn't see any evidence for host plant species prediction. I think "and host" should be deleted from the title.
  2. For lines 332 to 333, the authors mentioned 115 genomes used for lifestyle prediction in their Table 3. However, there were far fewer than 115 species in Table 3 but a lot of than 115 proteomes in this table. The total number does not match that in Figure 7. Some of the lines in table 3 represent genera instead of species and many of these genera have species with different lifestyles. For example, there are 34 proteomes in the genus Aspergillus and they were all classified as S (saprophytes). However, some species in this genus are necrotrophs and opportunistic pathogens.
  3. In the study by Hane et al. (2020) on fungal trophic classification/prediction that the authors cited, seven trophic types were identified among filamentous plant fungal pathogens using CAZyme profiles. With more types (7 vs 4), the predictive accuracy will understandably decrease. I think for a fair comparison, the authors should use the CAZyme profiles to classify the Oomycetes (similar to what Hane et al. did for filamentous plant fungal pathogens) and then compare the CAZyme results with those described in the current paper. 

Author Response

We greatly appreciate your comments and constructive feedback. The responses to your three remarks are as follows:

Point 1: The title is a bit misleading. While the paper demonstrated the predictive power of the positive selection data and genome properties for the four lifestyles of oomycetes, I didn't see any evidence for host plant species prediction. I think "and host" should be deleted from the title.

Response 1: We agree with this suggestion and we have modified the title accordingly.

Point 2: For lines 332 to 333, the authors mentioned 115 genomes used for lifestyle prediction in their Table 3. However, there were far fewer than 115 species in Table 3 but a lot of than 115 proteomes in this table. The total number does not match that in Figure 7. Some of the lines in table 3 represent genera instead of species and many of these genera have species with different lifestyles. For example, there are 34 proteomes in the genus Aspergillus and they were all classified as S (saprophytes). However, some species in this genus are necrotrophs and opportunistic pathogens.

Response 2: We have rephrased this section to make clearer that we have used 324 genomes from 115 species in total, as well as the legend from Figure 7, to highlight that the 64 genomes represented in the image corresponding to the validation are only a subset of the complete dataset. Additionally, we have added the species information for all the proteomes used to Table 8 for clarification.

Point 3: In the study by Hane et al. (2020) on fungal trophic classification/prediction that the authors cited, seven trophic types were identified among filamentous plant fungal pathogens using CAZyme profiles. With more types (7 vs 4), the predictive accuracy will understandably decrease. I think for a fair comparison, the authors should use the CAZyme profiles to classify the Oomycetes (similar to what Hane et al. did for filamentous plant fungal pathogens) and then compare the CAZyme results with those described in the current paper. 

Response 3: As suggested, we have constructed another model by using instead only CAZymes annotated from the same dataset of proteomes using the same methods and parameters for better comparison. We have included this model in the manuscript and made it available on the GitHub page. We have also rephrased this section to make clearer that the models have different number of prediction classes, and this may explain the higher observed accuracies for our models.

Round 2

Reviewer 1 Report

The authors have made several important revisions to the original manuscript that appears in a good form for publication. Yet, the organization of the figures may be improved.

Author Response

We thank you for pointing out that there is a lack of clarity due to the arrangement and presentation of our figures. We have improved this and made several changes to the organization of the figures, including a grouping of figures that are related in their topic.

  • We have added old figures 12 and 13 to figure 2.
  • We have added old figures 6 and  17 to figure 5 (now figure 4).
  • We have added old figure 15 to figure 8 (now figure 7).
  • We have moved old figure 4 to the supplementary figures (now figure 11).
  • We have moved old figure 16 from the supplementary to the body of the manuscript (now figure 5).

We are grateful for your valuable comments that have improved our manuscript a lot.

Reviewer 3 Report

The authors have addressed my concerns.

Author Response

We are thankful for your valuable comments that have helped significantly improve our manuscript.